# Relationship of the Difficulty of *Helicobacter pylori* Eradication with Drinking Habits and Allergic Disease

**DOI:** 10.3390/microorganisms10051029

**Published:** 2022-05-15

**Authors:** Kayoko Ozeki, Takahisa Furuta, Kazuhiro Hada, Yoshifumi Wakiya, Toshiyuki Ojima

**Affiliations:** 1Laboratory of Pharmacy Practice and Sciences, Aichi Gakuin University, Nagoya 4648650, Aichi, Japan; khada@dpc.agu.ac.jp (K.H.); y-wakiya@dpc.agu.ac.jp (Y.W.); 2Department of Community Health and Preventive Medicine, Hamamatsu University School of Medicine, Hamamatsu 4313192, Shizuoka, Japan; ojima@hama-med.ac.jp; 3Center for Clinical Research, Hamamatsu University School of Medicine, Hamamatsu 4313192, Shizuoka, Japan; furuta@hama-med.ac.jp

**Keywords:** *Helicobacter pylori*, eradication, immunoglobulin E levels, alcohol

## Abstract

Eradication of *Helicobacter pylori* (*H. pylori*) is crucial to reduce the risk of developing gastric ulcers and gastric cancer. Although immunoglobulin E (IgE) levels and alcohol consumption have been shown to influence the failure of *H. pylori* eradication, the relationship between these factors and the mechanism of failure has not been clarified. Because high IgE levels are associated with eradication failure, the purpose of this study was to clarify the factors leading to high IgE levels. Completed questionnaires and blood test data were collected from patients who visited a university hospital for *H. pylori* eradication. Logistic regression analysis was per-formed to examine the relationship between high IgE levels and allergic diseases. We also examined the relationship between alcohol intake and high IgE levels. Linear regression analysis was performed on the relationship between the amount of alcohol consumed and IgE measurements. The results showed that patients with allergic diseases and those with high alcohol intake had significantly higher IgE levels. High IgE levels are a risk factor for failure of *H. pylori* eradication that is associated with drinking habits and alcohol consumption, and our results suggest that daily alcohol consumption should be avoided even in non-allergic patients.

## 1. Introduction

*Helicobacter pylori* can cause gastric ulcers and gastric cancer [1,2,3,4]. Its eradication is very important to reduce the development of disease in *H. pylori* carriers.

The success rate of the temporary eradication of *H. pylori* has increased dramatically since 2015, when vonoprazan (a potassium-competitive acid blocker [P-CAB]) replaced proton pump inhibitors as a component of the eradication regimen [5]. However, eradication fails in some people.

We previously found that eradication failure is associated with hay fever in 2016 [6], with alcohol consumption in female patients in 2019 [7], and with high IgE levels taking into account antimicrobial susceptibility in 2021 [8]. Previous studies have thus shown that pollinosis, IgE levels, and alcohol consumption each affect the failure of *H. pylori* eradication, but the relationship between these factors and their mechanisms has not been clarified.

In this study, because high IgE levels are associated with bacteria eradication failure [8], we aimed to clarify the factors causing high IgE levels. In particular, we focused on patients with allergic diseases and their drinking habits, considering the possibility that a history of allergic diseases [6] and alcohol consumption [7] may be related, according to our previous work.

## 2. Materials & Methods

### 2.1. Subjects and Study Procedure

Completed questionnaires and blood test data of 250 patients who visited the outpatient unit specialized for *H. pylori* in Hamamatsu University Hospital for *H. pylori* eradication between April 2017 and December 2020 were obtained and analyzed; the patients were supplied with a written explanation of the study aims, and consent was obtained. Questionnaire items included age, sex, allergies (hay fever, rash, asthma, atopic dermatitis, and other allergic diseases), alcohol consumption, smoking, previous eradication experience, and number of eradications performed thus far. The total IgE level (unit: IU/mL) was measured as a blood test with fluorescence enzyme immunoassay (FEIA; one of the EIA methods, in which an antigen–antibody reaction is performed using antigens or antibodies labeled with enzymes, and the fluorescence intensity is measured; IgE reference value, 173 IU/mL). The exclusion criterion was failure to respond to the required items.

The drugs amoxicillin, clarithromycin, and vonoprazan were used for primary eradication, whereas amoxicillin, metronidazole, and vonoprazan were used for secondary eradication. After tertiary eradication, the choice of treatment was left largely to the discretion of the physician, but in many cases, sitafloxacin and minocycline were used instead of amoxicillin.

This study was approved by the Medical Ethics Committee of Hamamatsu University School of Medicine (No. 17-072).

### 2.2. Statistical Analysis

We performed a simple tabulation of attributes such as sex, age, and number of *H. pylori* eradications. We also calculated the geometric mean of the IgE data for each item.

Next, logistic regression analysis was conducted to examine the relationship of high IgE levels (>173 IU/mL) as the dependent variable with the presence or absence of various allergic diseases (hay fever, rash, asthma, atopic dermatitis) and the presence or absence of the allergic diseases as the independent variables.

In Model 1, the presence or absence of various and all allergic diseases was used as an independent variable; in Model 2, age and sex were added as covariates; and in Model 3, the presence or absence of alcohol consumption was added to age and sex.

When the presence of alcohol consumption was added to the covariates, the association between the presence of alcohol consumption and high IgE levels was significant for all allergic diseases. Thus, logistic regression analysis was conducted separately to examine the association between the presence of alcohol consumption and high and low IgE levels. In Model 1, we used the presence of alcohol consumption as an independent variable; in Model 2, we added sex and age as covariates; and in Model 3, we added sex, age, smoking status, and medication status.

In addition, taking into account the possibility of confounding by allergic diseases, we stratified the analysis by the presence or absence of allergic diseases to identify the relationship between alcohol consumption and IgE.

The amount of alcohol in grams per day was determined from the volume of alcohol consumed, and a linear regression analysis was performed on the relationship with IgE measurements. In Model 1, the independent variable was grams of alcohol per day; in Model 2, sex and age were added as covariates; and in Model 3, sex, age, smoking status, and medication status were added.

Statistical analysis was performed with JMP13.

## 3. Results

The patient demographics of the study are shown in Table 1. Most of the patients were aged 61 years or older, and the number of patients currently taking medication for eradication was high at over 60%. We also obtained answers about the patients’ lifestyle habits, such as drinking and smoking, as well as the presence of allergic diseases. About 80% of the patients had been tested for *H. pylori*, and 60% had been treated for eradication. In addition, about 10% of the patients had undergone three or more eradication treatments.

The geometric mean of IgE levels in all patients was 8.12. Patients with a geometric mean higher than 9 were more likely to be male, drink alcohol, have allergic diseases, and have undergone more than three *H. pylori* eradication attempts.

Table 2 presents the results of the logistic regression analysis of the relationship between the presence of various allergic diseases and IgE levels. In this table, IgE levels are divided into standard—less than 173 IU/mL—and high—173 IU/mL or more—and the odds ratios of having various diseases with a high IgE level are shown. In Models 1 and 2, IgE levels were significantly higher in patients with hay fever, rash, asthma, and atopy. In Model 3, in which alcohol consumption was added as a covariate, IgE levels were significantly higher in patients with rash and asthma. Characteristically, “drinking alcohol” was associated with significantly higher IgE levels in patients with these four allergic diseases.

Table 2 also shows the results of the relationship between the presence of all allergic diseases (any allergic disease or multiple allergic diseases, including hay fever, rash, asthma, and atopy) and high IgE levels (IgE ≥ 173 IU/mL) collected from patients. In Models 1, 2, and 3, patients with allergic diseases had significantly higher IgE levels. Presence of alcohol consumption” was also associated with significantly higher IgE in Model 3.

We also added smoking status and medication status as covariates to Model 3 in Table 2, and the results were similar to those of Model 3.

Based on these results, we analyzed the association between the presence of alcohol consumption and high IgE levels (>173 IU/mL) by logistic regression (Table 3). The odds ratio of patients with alcohol consumption to those without alcohol consumption was significantly higher for a high IgE level in all models (Models 1, 2, and 3).

Table 4 shows the relationship between the presence of alcohol consumption and high IgE levels, stratified by allergic disease. In Model 1, alcohol consumption was significantly associated with higher IgE levels, regardless of the presence or absence of allergy. In Models 2 and 3, patients without allergy and with alcohol consumption were significantly more likely to have higher IgE levels.

In addition, Table 5 shows the results of linear regression analysis of the association between daily alcohol intake and IgE. In all of the models (Models 1, 2, and 3), significantly higher alcohol intake was clearly associated with higher IgE.

## 4. Discussion

To our knowledge, this is the first study to examine the relationship between high IgE levels in patients with allergic diseases and their drinking habits, using patients who struggled to eradicate *H. pylori*. The results identified a relationship between high IgE levels and patients’ drinking habits and the amount of alcohol consumed. Therefore, patients who tend to drink more frequently and with high alcohol consumption have higher IgE levels and are more likely to experience eradication failure, suggesting that multiple eradication attempts may be necessary.

The patients included in this study had visited university hospitals and were considered to be a difficult group in terms of *H. pylori* eradication because they had previously experienced eradication failure at general practitioners or private hospitals. These characteristics suggest that it would be somewhat easier to find the involvement of patient attributes regarding the difficulty of eradication than in the general population.

The results of this study showed that drinking habits were associated with high IgE levels. We considered IgE levels exceeding 173 IU/mL to be high because the supplier of the IgE test kits set a standard value of 173 U/mL or lower based on previous studies of the IgE distribution in healthy individuals and allergic patients [9].

According to previous research, an increase in IgE-containing plasma cells results in various non-specific chronic inflammatory disorders of the upper gastrointestinal tract [10]. In addition, the number of IgE-positive cells was significantly elevated in patients with chronic gastritis compared to normal mucosa, and patients with *H. pylori*-associated gastritis had a marked accumulation of IgE-positive plasma cells, suggesting that IgE-mediated immune responses play an important role in the development of *H. pylori*-associated gastritis [11,12]. A previous allergy-related study showed that *H. pylori*-infected stomachs with ulcers have inadequate regulatory T cell responses, and it has been hypothesized that regulatory T cell-mediated suppression of inflammation and the bactericidal activity of epithelial cells might affect the maintenance of increased *H. pylori* density and long-term bacterial colony formation [13]. Although regulatory T cells prevent allergy by suppressing T2 cells [14], they may also be involved in IgE production by altering regulatory T cell responses [15]. Considering the above hypotheses, people with high IgE levels may have a high density of *H. pylori* and prolonged bacterial colony formation; previous studies examining the relationship between the rate of *H. pylori* eradication and *H. pylori* density found that the higher the bacterial density of *H. pylori*, the significantly more difficult the eradication [16].

Regarding alcohol consumption, clarithromycin–an antibiotic contained in the eradication agent–is known to be metabolized mainly by CYP3A4 [17], and induction in cytochrome P450 including CYP3A4 is known to occur in response to regular, heavy alcohol consumption [18]. The effect of alcohol consumption has also been reported on elevated serum IgE levels [19,20,21]. Although some of the mechanisms remain unknown, alcohol consumption has been suggested to affect the immune system and cause complex changes in cytokines, including IgE production [22]. Alcohol intake has been linked to decreased type 1 helper T cell (Th1)-related cytokine levels and responses as well as increased type 2 helper T cell (Th2)-related cytokine levels [23]. Cytokines are mediators of the balance between Th1 and Th2 immune responses, and elevated Th2 cytokine levels increase IgE production [24]. In addition, alcohol consumption promotes the absorption of lipopolysaccharides from the intestine, and lipopolysaccharides appear to play a role in the immune system changes caused by alcohol abuse, increasing IgE production under certain circumstances [25]. It has also been reported that acetaldehyde, a metabolite of alcohol, induces histamine release from human lung mast cells and the degranulation of mast cells [26].

Although the success rate of eradication has increased in recent years, some people still struggle to eradicate *H. pylori*, and this may be due to patient factors. In a previous study, we found that patients with high IgE levels were less likely to achieve eradication, even after taking into account their sensitivity to the antimicrobial agents used for eradication [8]. This study suggested that patients with allergic diseases may be more likely to experience eradication failure because they tend to have higher IgE levels.

Although there have been few similar studies in Asian patients regarding the association between alcohol consumption and IgE, the present study showed results consistent with previous findings of a significant association between alcohol consumption and high IgE levels [19,20,21,25].

In addition, an analysis stratified by the presence or absence of allergic disease revealed that the association between alcohol consumption and higher IgE levels was more apparent in non-allergic patients than in those with allergic disease. A previous study found similar results, and its authors speculated that this may be because allergic disease itself is a stronger determinant of elevated IgE and that it may therefore mask some of the effects of alcohol consumption [27].

We also found that the IgE level increased as the amount of alcohol consumed increased. Some studies have specifically evaluated the possible effects of regular alcohol consumption on total serum IgE levels, and the results showed that alcohol abusers have higher total IgE levels than abstainers [19,25]. These results suggest that excessive alcohol consumption may affect IgE levels and hamper *H. pylori* eradication.

There are five main limitations of this study. First, the presence of various allergic diseases was self-reported by the patients. However, hay fever, for example, is a disease that patients are most aware of because symptoms begin to appear at the same time every year and ameliorate over time. Asthma is characterized by coughing and breathlessness and is often already diagnosed by doctors. Rashes and atopic dermatitis are also diseases that are readily evident and can be easily recognized by the patients themselves. Nonetheless, we were unable to obtain information on the severity of allergic symptoms. Second, the frequency and amount of alcohol consumption was also self-reported by the patients. However, the questionnaire was based on the National Health and Nutrition Survey and was sufficiently detailed to determine the consumption of alcohol in terms of both frequency and quantity. Third, as we have mentioned before, the target population is biased because they are patients of a university hospital. However, because the eradication rate of P-CABs has increased considerably in recent years [5], this fact may be considered a strength of this study. Fourth, because this is an epidemiological study, we were unable to examine the molecular basis of the study. Fifth, patient-specific IgE values were not obtained in this study. This must be discussed further in future studies.

IgE has a short half-life of two to seven days, and a cohort study of alcoholics reported a fall in the total serum IgE after abstinence from alcohol [28,29,30]. Our results are thus worth considering because they might help patients to reevaluate their drinking habits.

Regarding the drinking habit- and alcohol consumption-related elevation in IgE, the results suggest that daily alcohol consumption should be avoided, even as a risk factor for failure of *H. pylori* eradication.

## 5. Conclusions

Our analysis shows that the association between alcohol consumption and higher IgE levels was more apparent in non-allergic patients than in those with allergic disease. Given the association between high alcohol consumption and high IgE levels, as well as the difficulty of *H. pylori* eradication, it might be beneficial for both allergic and non-allergic patients to avoid drinking alcohol on a daily basis.

## Figures and Tables

**Table 1 microorganisms-10-01029-t001:** Patients’ demographic characteristics.

Characteristic	*n*	%	Geometric Average of IgE
**Sex**			
Male	109	43.6	**9.68**
Female	141	56.4	6.91
**Age, years**			
20–40	40	16.0	8.32
41–60	82	32.8	8.72
≥61	128	51.2	7.67
**Medication status**			
Taking medication	155	62.0	8.18
No medication	78	31.2	8.33
**Smoking status**			
Smoking (+)	23	9.2	8.71
Smoking (−)	211	84.4	8.19
**Drinking status**			
Drinking (+)	127	50.8	**9.69**
Drinking (−)	102	40.8	6.66
**History of allergic diseases**			
Rash	30	12.0	**9.86**
Asthma	15	6.0	**15.88**
Atopic dermatitis	8	3.2	**17.63**
Hay fever	92	36.8	**9.27**
Allergic diseases of any kind (+)	116	46.4	**9.73**
Allergic diseases of any kind (−)	134	53.6	6.72
**Inspection of *H. pylori***			
Inspection (+)	195	78.0	8.58
Inspection (−)	33	13.2	6.83
**Eradication of *H. pylori***			
Eradication (+)	153	61.2	8.59
Eradication (−)	70	28.0	7.76
**Number of *H. pylori* eradication treatments**			
One or two	125	50.0	8.06
Three or more	26	10.4	**11.82**

**Table 2 microorganisms-10-01029-t002:** Association between the presence of various allergic diseases and a high IgE level (≥173 IU/mL).

	Model 1			Model 2			Model 3		
	OR	95%CI	*p*-Value	OR	95%CI	*p*-Value	OR	95%CI	*p*-Value
**Hay fever (+)**	**1.85**	1.04–3.30	**0.036**	**1.88**	1.01–3.50	**0.047**	1.47	0.77–2.82	0.239
**Alcohol (+)**							**2.61**	1.32–5.16	**0.006**
**Rash (+)**	**2.47**	1.13–5.42	**0.024**	**2.92**	1.27–6.70	**0.012**	**2.56**	1.09–6.01	**0.032**
**Alcohol (+)**							**2.76**	1.40–5.46	**0.004**
**Asthma (+)**	**4.89**	1.66–14.86	**0.004**	**6.89**	2.19–21.62	**<0.001**	**6.38**	1.98–22.28	**0.002**
**Alcohol (+)**							**2.86**	1.45–5.87	**0.002**
**Atopic dermatitis (+)**	**5.06**	1.21–25.23	**0.027**	**5.89**	1.29–29.60	**0.022**	4.17	0.91–22.48	0.065
**Alcohol (+)**							**2.60**	1.33–5.23	**0.005**
**Allergic diseases of any kind (+)**	**3.00**	1.66–5.42	**<0.0** **01**	**3.43**	1.81–6.54	**<0.001**	**2.76**	1.42–5.39	**0.003**
**Alcohol (+)**							**2.43**	1.22–4.84	**0.012**

Analysis method: Logistic regression analysis. Notes: Model 2 was adjusted for age and sex. Model 3 was adjusted for sex, age, and drinking status. Bold text, statistically significant at *p* < 0.05. Abbreviations: OR, odds ratio; CI, confidence interval.

**Table 3 microorganisms-10-01029-t003:** Association between alcohol consumption and a high IgE level (≥173 IU/mL).

	Model 1			Model 2			Model 3		
	OR	95%CI	*p*-Value	OR	95%CI	*p*-Value	OR	95%CI	*p*-Value
**Alcohol (+)**	**3.27**	1.75–6.36	**<0.001**	**2.78**	1.45–5.45	**0.003**	**2.70**	1.36–5.34	**0.004**

Analysis method: Logistic regression analysis. Notes: Model 2 was adjusted for age and sex. Model 3 was adjusted for age, sex, smoking status, and medication status. Bold text, statistically significant at *p* < 0.05. Abbreviations: OR, odds ratio; CI, confidence interval.

**Table 4 microorganisms-10-01029-t004:** Association between alcohol consumption and a high IgE level by the presence or absence of allergic disease (IgE ≥ 173 IU/mL).

	Model 1			Model 2			Model 3		
	OR	95%CI	*p*-Value	OR	95%CI	*p*-Value	OR	95%CI	*p*-Value
**Allergic disease (−)**	**3.78**	1.35–10.64	**0.012**	**3.10**	1.07–8.97	**0.037**	**3.11**	1.06–9.12	**0.038**
**Allergic disease (+)**	**2.36**	1.03–5.71	**0.042**	2.11	0.84–5.29	0.111	2.22	0.86–5.73	0.101

Analysis method: Logistic regression analysis. Notes: Model 2 was adjusted for age and sex. Model 3 was adjusted for age, sex, smoking status, and medication status. Bold text, statistically significant at *p* < 0.05. Abbreviations: OR, odds ratio; CI, confidence interval.

**Table 5 microorganisms-10-01029-t005:** Association between daily alcohol intake and IgE measurements.

	Model 1		Model 2		Model 3	
	β	*p*-Value	β	*p*-Value	β	*p*-Value
**Daily alcohol intake (g)**	**0.202**	**0.036**	**0.204**	**0.045**	**0.218**	**0.034**

Analysis method: Linear regression analysis. Notes: Model 2 was adjusted for sex and age. Model 3 was adjusted for sex, age, smoking status, and medication status. β, standard regression coefficient. Bold text, statistically significant at *p* < 0.05.

## Data Availability

Not applicable.

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
