# Peer review of "Relationship of the Difficulty of Helicobacter pylori Eradication with Drinking Habits and Allergic Disease"

_microorganisms, 2022, doi:10.3390/microorganisms10051029_

Round 1
Reviewer 1 Report
The present work shows interesting associations between different factors more or less related to a failure in the H. pylori eradication. However, some important informations are lacking on 1) the specific IgE levels (total IgE is a too crude parameter), 2) the degree of alcohol consumption, 3) the atopic/asthma status and 4) the previous drug treatment used to eradicate H. pylori from patients, to ascertain/refine the observed associations. In addition, a deeper discussion of the possible (molecular) mechanisms sustaining the relationships between the different factors should be presented.
Author Response
We are grateful to you for your critical comments and valuable suggestions. We have revised our paper in accordance with these comments, and our responses are listed below in a point-by-point fashion.
Reviewer 1
We are grateful to you for your critical comments and valuable suggestions. We have revised our paper in accordance with these comments, and our responses are listed below in a point-by-point fashion.
【Comments and Suggestions for Authors】
The present work shows interesting associations between different factors more or less related to a failure in the H. pylori eradication.
→Thank you for your comments.
However, some important informations are lacking on 1) the specific IgE levels (total IgE is a too crude parameter), 2) the degree of alcohol consumption, 3) the atopic/asthma status and 4) the previous drug treatment used to eradicate H. pylori from patients, to ascertain/refine the observed associations. In addition, a deeper discussion of the possible (molecular) mechanisms sustaining the relationships between the different factors should be presented.
→Thank you for pointing those out.
- As for 1), we were unable to obtain information from the patient, so we noted that in the limitation (on line 244-245).
- As for 2), the patient's self-reported alcohol intake was obtained (green highlighted area on line 54) and noted in the method. We also noted that it is self-reported in the limitation.
- As for 3), we could not ascertain the extent of the allergy, so we noted it in the limitation (on line 236).
- As for 4), the names of the drugs used for primary, secondary, and tertiary or more eradication are listed in the method (green highlighted area on line 61-65), and we have published a paper about IgE and the difficulty of eradication, taking into account antimicrobial susceptibility. Therefore, we added a note on line 38 regarding antimicrobial susceptibility.
- As for molecular mechanisms, since this is an epidemiological study, it is difficult to examine the molecular basis for the study, so we have included that in the limitation (on line 243-244).
Reviewer 2 Report
In this paper entitled “Relationship of the Difficulty of Helicobacter pylori Eradication with Drinking Habits and Allergic Disease”, the authors showed the association between high alcohol consumption and high IgE levels, as well as the difficulty of H. pylori eradication.
Although the data are informative, the meaning of observable facts has yet to be fully elucidated.
There are serious concerns that need to be addressed as follows.
Major Concerns
[1] Except IgE levels, what other factors are influenced by excessive alcohol consumption and hamper H. pylori eradication ?
[2] I would suggest the authors to elaborately discuss the most crucial point “What is the molecular basis for the association between high IgE levels and H. pylori eradication failure?”.
Author Response
We are grateful to you for your critical comments and valuable suggestions. We have revised our paper in accordance with these comments, and our responses are listed below in a point-by-point fashion.
Reviewer 2
We are grateful to you for your critical comments and valuable suggestions. We have revised our paper in accordance with these comments, and our responses are listed below in a point-by-point fashion.
【Comments and Suggestions for Authors】
In this paper entitled “Relationship of the Difficulty of Helicobacter pylori Eradication with Drinking Habits and Allergic Disease”, the authors showed the association between high alcohol consumption and high IgE levels, as well as the difficulty of H. pylori eradication.
→Thank you for your comments.
Although the data are informative, the meaning of observable facts has yet to be fully elucidated.
There are serious concerns that need to be addressed as follows.
Major Concerns
[1] Except IgE levels, what other factors are influenced by excessive alcohol consumption and hamper H. pylori eradication ?
[2] I would suggest the authors to elaborately discuss the most crucial point “What is the molecular basis for the association between high IgE levels and H. pylori eradication failure?”.
→Thank you for pointing those out.
- As for [1], We have added the following text to the Discussion section: “Regarding alcohol consumption, clarithromycin, an antibiotic contained in the eradication agent, is known to be metabolized mainly by CYP3A4 [17], and also induction in cytochrome P450 including CYP3A4 is known to occur in response to regular, heavy alcohol consumption [18].” (on line 193-196).
- As for [2], since this is an epidemiological study, it is difficult to examine the molecular basis for the study, so we have included that in the limitation (on line 243-244). However, we have added more information to the Discussion section: “According to previous research, an increase in IgE-containing plasma cells results in various non-specific chronic inflammatory disorders of the upper gastrointestinal tract [10]. In addition, the number of IgE-positive cells was significantly elevated in patients with chronic gastritis compared to normal mucosa, and patients with H. pylo-ri-associated gastritis had a marked accumulation of IgE-positive plasma cells, sug-gesting that IgE-mediated immune responses play an important role in the develop-ment of H. pylori-associated gastritis [11-12].” (on line 176-182).
Round 2
Reviewer 1 Report
No comments.
Reviewer 2 Report
The paper is suitable for publication in this Journal.